# Prevalence, Incidence, and Risk Factors for Intestinal Colonization Due to Fluoroquinolone-Resistant ST131 *Escherichia coli*: a Longitudinal Study in Highly Dependent, Long-Term Care Facility Residents

Elena Salamanca-Rivera,[a,b,e] Lorena López-Cerero,[a,b,c,e] Jose Manuel Rodríguez-Martínez,[b,c,e] Alvaro Pascual,[a,b,c,e] Jesús Rodríguez-Baño[a,b,d,e]

aUnidad Clínica de Enfermedades Infecciosas y Microbiología, Hospital Universitario Virgen Macarena, Seville, Spain
bInstituto de Biomedicina de Sevilla and CSIC, Seville, Spain
cDepartamento de Microbiología, Universidad de Sevilla, Seville, Spain
dDepartamento de Medicina, Universidad de Sevilla, Seville, Spain
eCentro de Investigación en Red en Enfermedades Infecciosas (CIBERINFEC), Madrid, Spain

Elena Salamanca-Rivera and Lorena López-Cerero contributed equally to this study. Author order was determined by agreement of authors.
Alvaro Pascual and Jesús Rodríguez-Baño contributed equally to this study as senior authors.

**ABSTRACT** *Escherichia coli* ST131 clade C is an important driver for fluoroquinolone resistance (FQ-R). We conducted a prospective observational study in residents from two long-term care facilities (LTCFs) in Seville, Spain, in 2018. Fecal swabs and environmental samples were obtained. *E. coli* isolates were screened for clade C, FQ-R ST131 by PCR, and molecular typing by PFGE; representatives from pulsotypes were studied by whole-genome-sequencing (WGS) and assigned to lineages (cgSTs). Prevalence of colonization at each time point, incidence density, and risk factors for acquisition were studied. Seventy-six FQ-R ST131 *E. coli* isolates belonging to 34 cgSTs were obtained; 24 belonging to subclade C1 (116 isolates, 65.9%) and 10 to C2 (60, 34.1%). C1 lineages showed lower virulence scores than C2 (median [IQR], 19 [18 to 20] versus 21 [20 to 21.5], $P = 0.001$) and higher number of plasmids (4 [3 to 5] versus 2 [2 to 3], $P = 0.01$). $aac(6')$-$lb$-$cr$ and $bla_{OXA-1}$ were less frequent in C1 than C2 (2 [8.3%] versus 6 [60%], $P = 0.003$ for both); ESBL genes were detected in eight (33.3%) C1 (5 $bla_{CTX-M-27}$) and three (30%) C2 (all $bla_{CTX-M-15}$). Of the 82 residents studied, 49 were colonized at some point (59.7%), with a pooled prevalence of 38.6%. Incidence density of new lineage acquisition was 2.22 per 100 resident weeks (1.28 and 0.93 C1 and C2 subclades, respectively). Independent risk factors for acquisitions were having a colonized roommate (HR = 4.21; 95% CI = 1.71 to 10.36; $P = 0.002$) and urinary or fecal incontinence (HR = 2.82; 95% CI = 1.21 to 6.56; $P = 0.01$). LTCFs are important reservoirs of clade C ST131 *E. coli*. The risk factors found suggest that cross-transmission is the most relevant transmission mechanisms.

**IMPORTANCE** We aimed at investigating the microbiological and epidemiological features of clade C fluoroquinolone-resistant ST131 *E. coli* isolates colonizing highly dependent residents in long-term care facilities (LTCFs) during 40 weeks and the risk factors of acquisition. Isolates from C1 and C2 subclades were characterized in this environment. The clonality of the isolates was characterized and they were assigned to lineages (cgSTs), Resistance genes, virulence factors, and plasmids were also described. This study suggests that cross-transmission is the most relevant transmission mechanisms; however, environmental colonization might also play a role. We believe the data provide useful information to depict the epidemiology of these bacteria by merging detailed microbiological and epidemiological information.

Address correspondence to Jesús Rodríguez-Baño, jesusrb@us.es.

The authors declare no conflict of interest.

**KEYWORDS** *Escherichia coli*, ST131, fluorquinolone resistance, multidrug resistance, risk factors

*E*scherichia coli* is a very frequent human pathogen, being the most frequent cause of complicated and uncomplicated urinary tract infections, among other types of infections. For this reason, antimicrobial resistance in *E. coli* has important consequences, and understanding the clonal and clinical epidemiology of resistant isolates is important to design control measures.

During the last decades, a significant increase in resistance rates to some first-line drugs for the treatment of invasive *E. coli* infections (such as cephalosporins and fluoroquinolones) has been partly linked to the successful spread of the clonal complex ST131 (1). Three clades have been distinguished within this complex, clade C being the most prevalent worldwide. This clade is associated with high-level fluoroquinolone resistance and in fact is considered the main driver for the rapid spread of worldwide resistance to these drugs in *E. coli* (2). In addition, clade C is differentiated into two distinct subgroups or subclasses, C1 and C2. The latter, named C2/H30Rx, has been associated with the acquisition and spread of cephalosporin-resistance mediated by some extended-spectrum $\beta$-lactamases (ESBL), mostly CTX-M-15 (1). The subclade C1/H30R was initially not associated with ESBL production; however, a distinct lineage of C1 subclade, associated with the production of CTX-M-27, was described in Japan (3) and has also been detected in Canada and some European countries (4–6), recognized recently as a new global successful ESBL-producing ST131 subgroup.

Most studies on the clinical epidemiology of ST131 have been performed on ESBL-producers; however, a high proportion of fluoroquinolone-resistant (FQ-R) ST131 isolates do not produce ESBLs (1, 7) despite their potential epidemiological importance. Recently, a higher frequency of C1/CTX-M-27 compared with C2/CTX-M-15 was observed in two long-term care facilities (LCTF) in Seville, Spain (8). In this context, we studied the prevalence, incidence, and risk factors for the colonization with FQ-R (as a marker for C clade isolates) ST131 *E. coli* in residents of these LCTFs over two 12-week periods, with the intention to provide a comprehensive view of the epidemiology of these isolates in this epidemiological context, and specifically, of the emerging C1 subclade producing CTX-M-27.

## RESULTS

**Participants.** Overall, 82 residents, 52 from LTCF-1 and 30 from LTCF-2 (62% of bed occupation at starting time point), were included. Because of the similarity in structure, features of the residents and incidence of FQ-R ST131 *E. coli* acquisition, the data for both LTCF were merged. The median age of the patients was 83 years and 62.1% were female. More than 90% needed assistance for basic hygiene and toilet use. These and other baseline features and comorbidities are shown in Table 1. During follow-up, five residents left the LTCFs, eight were admitted to an acute care hospital, seven died, and the relatives of another withdraw the consent for continuing in the study. Therefore, the number of residents from whom samples were taken at the different time points decreased from 82 to 34. Their median follow-up time was 31 weeks (IQR, 13 to 43). Overall, 459 samples were taken over the complete study period.

**Features of the FQ-R ST131 *E. coli* isolates.** Overall, 176 FQ-R ST131 *E. coli* isolates belonging to 34 cgSTs were obtained from 49 residents (Fig. 1). Of these, 24 strains belong to subclade C1 (116 isolates, 65.9% of all FQ-R ST131 *E. coli*) and 10 to subclade C2 (60 isolates, 34.1%).

The antimicrobial resistance, resistance genes, plasmid, and virulence profiles of all lineages are shown in Table 2. Overall, C1 isolates had lower virulence score than C2 (median [IQR], 19 [18 to 20] versus 21 [20 to 21.5], $P = 0.001$) and higher number of plasmids (four [3 to 5] versus two [2 to 3], $P = 0.01$). Using a $P < 0.01$ threshold because of multiple comparisons, no significant differences in resistance to the target antimicrobials tested were found between lineages belonging to C1 and C2. Regarding resistance genes, *aac(6')-lb-cr* and $bla_{OXA-1}$ were less frequent in C1 than C2 lineages (two [8.3%] versus six [60%], $P = 0.003$ for

**TABLE 1** Features of 82 residents in two long-term care facilities participating in the colonization study

| Variable | All patients[a] (n = 82) |
|---|---|
| Median age in yrs (IQR) | 83 (72 to 88) |
| Female gender | 51 (62.1) |
| Private bathroom | 30 (36.6) |
| Dependent for personal hygiene | 76 (92.7) |
| Dependent for toilet use | 75 (91.5) |
| Wheelchair use | 28 (34.1) |
| Urinary or fecal incontinence | 30 (36.6) |
| Diabetes mellitus | 17 (20.7) |
| Chronic peripheral vascular disease | 60 (73.2) |
| Chronic pulmonary disease | 12 (14.6) |
| Chronic renal insufficiency | 12 (14.6) |
| Chronic liver disease | 9 (11.0) |
| Hemiplegia | 8 (9.8) |
| Malignancy | 9 (11.0) |
| Unable to provide informed consent | 50 (70.0) |
| Urinary catheter | 5 (6.1) |
| Major surgery (previous yr) | 9 (11.0) |
| Hospitalization during previous yr | 13 (17.1) |
| Antibiotics use during follow-up | 45 (54.9) |

[a]Data are number of patients (percentage) except where specified.

both genes). IncFIB plasmids were more frequent in C1 (22 [91.7%] versus four [40%], $P = 0.003$), and regarding virulence genes, *afaA*, *afaC*, and *afaD* were less frequent (zero versus six [60%], $P < 0.001$ for the three genes), as was *nfaE* (two [8.3%] versus six [60%], $P = 0.003$). Regarding ESBL genes, they were detected in eight (33.3%) C1 strains (five $bla_{CTX-M-27}$ and one each of $bla_{CTX-M-14}$, $bla_{CTX-M-15}$, and $bla_{SHV-12}$) and three (30%) C2 strains (all $bla_{CTX-M-15}$). Only three of the five C1/CTX-M-27 harbored the M27PP1 phage region.

**Prevalence of colonization and incidence density of FQ-R ST131 acquisition.** Overall, 49 residents (59.7%) were found to be colonized with FQ-R ST131 *E. coli* at some point during follow-up. The prevalence of colonization at the different time points ranged from 32.3% to 48.8%, with a pooled prevalence of 38.6%. No trend for increase or reduction in prevalence during the study period was evident. The pooled prevalence of colonization with C1 and C2 isolates were 21.8% and 16.8%, respectively; overall, the prevalence with ESBL-producing FQ-R ST131 *E. coli* was 14.8%.

During the follow-up and excluding prevalent colonizations detected in the baseline sample, 34 residents among the 77 who had at least two samples had at least one acquisition of a new FQ-R ST131 *E. coli* strain (incidence density for a first acquisition, 1.67 residents per 100 resident weeks); because 11 residents acquired two different cgSTs during follow-up, there was a total of 45 acquisition episodes (incidence density of any new lineage acquisition, 2.22 per 100 resident weeks). Among these 45 acquisitions, 26 and 19 were of lineages belonging to C1 and C2 clades (57.8% and 42.2%), respectively (incidence density of C1 and C2 acquisitions, 1.28 and 0.93 per 100 resident weeks).

**Transmission episodes and environmental colonization.** Overall, 31 episodes of possible transmission caused by 13 strains were identified (seven belonging to C1, causing 14 transmissions, and six to C2, causing 17 transmissions). Overall, four strains caused 15 of these transmission episodes (48.3%) and were considered as highly transmitted; two belongs to C1 clade (cgST 131873 and cgST 162872) and the others to C2 (cgST 131563 and cgST131904). Although statistical analyses were not possible due to low numbers, no specific differential phenotypic or genotypic characteristics were evident for these lineages.

The average duration of colonization per lineage and patient was 11.9 weeks (SD 0.47). In 19 residents (38.7% of colonized residents) the duration of colonization was ≥12 weeks (long colonizers), and were caused by 14 of the 34 lineages (41.1%),

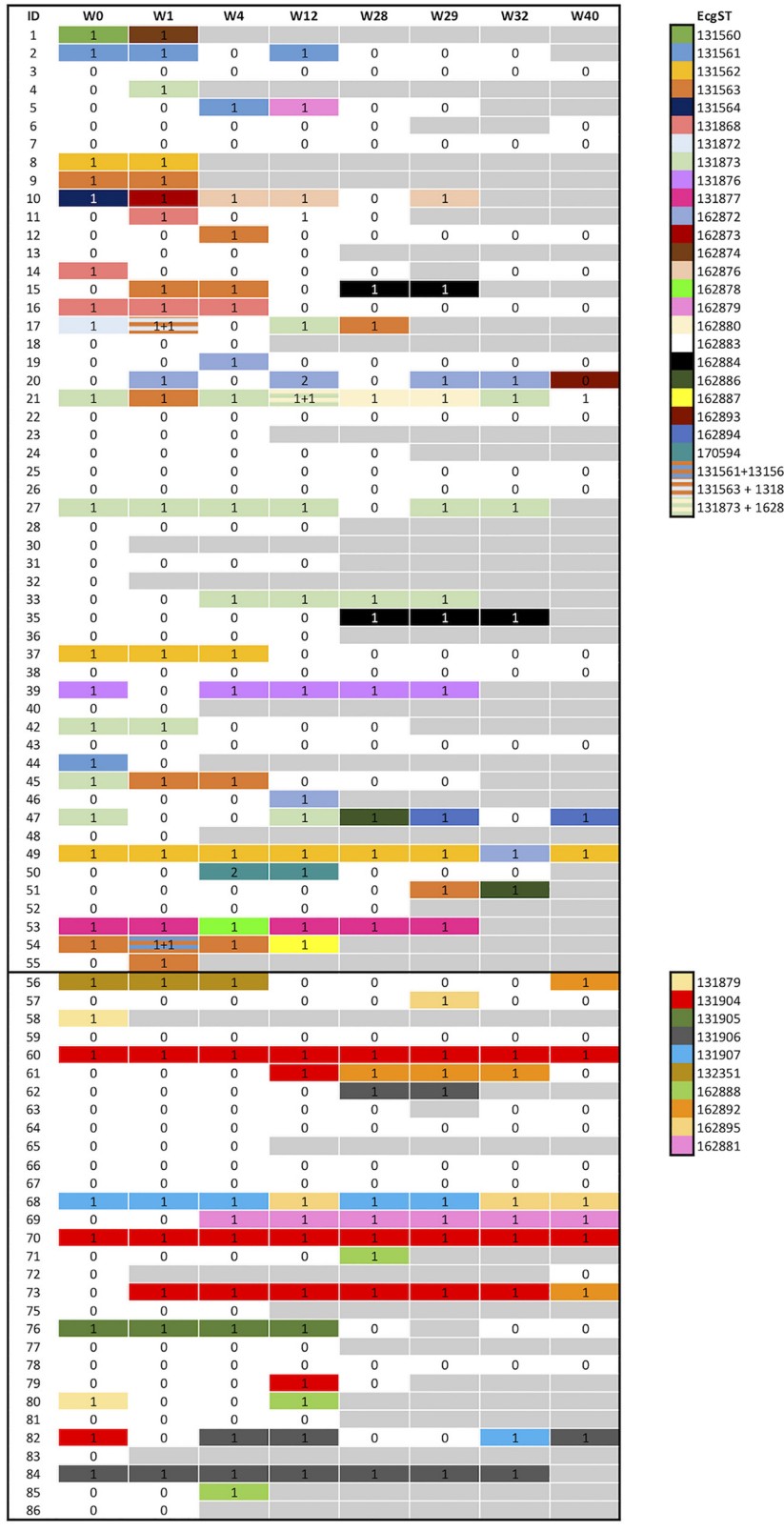

**FIG 1** Fluoroquinolone-resistant ST131 *E. coli* isolates obtained during the study. Each row is a participating resident (1 to 55, patients admitted to long-term care facility 1, and 56 to 86 patients admitted to long-term care facility 2), and each column is a time point for colonization study. Squares filled with a color and a number represent colonization with a lineage, and the number is the number of isolates; gray squares mean sample were not obtained at that specific time point.

**TABLE 2** Features of the lineages of FQ-R ST131 E. coli isolates

| Clade | Lineage according to cgST | No. of patients/ isolates | Resistance profile[a] | Pathogenicity islands | Target β-lactamase genes | Other resistance genes[b] | Plasmids[c] (other than IncF) | Virulence genes[c] (virulence score) |
|---|---|---|---|---|---|---|---|---|
| C1 | 131560 | 1/1 | TSX | aadA5, drf | | qnrS1 sul1, sul2 | IncI1, IncN | chuA, cia, fyuA, gad, iha, irp2, iss, iucC, iutA, kpsE, kpsII, papA, sat, sitA, traT, usp, yfcV (19) |
| | 131561[e] | 3/6 | | | | | | IncI1, Col156 | chuA, cia, fyuA, gad, iha, irp2, iss, iucC, iutA, kpsE, kpsII, papA, sat, sitA, traT, usp, yfcV (19) |
| | 131562[e] | 3/12 | AMC, GEN, TOB, TSX | | bla_CTX-M-1b | aac(3)-IId | IncI1, IncN, Col156, IncB/O/K/Z | chuA, cia, fyuA, gad, iha, irp2, iss, iucC, iutA, kpsE, kpsII, papA, sat, senB, sitA, traT, usp, yfcV (20) |
| | 131568 | 3/5 | | | bla_TEM-1b | | IncX4 | afaD, chuA, fyuA, gad, iha, irp2, iss, iucC, iutA, kpsE, kpsII, papA, sat, sitA, traT, usp, yfcV (19) |
| | 131873[d,e] | 8/21 | CAZ, FEP, TSX | aadA5, drf | bla_CTX-M-27 | aph(3'')-Ib, aph(6)-Id, sul1, sul2, tetA | Col156 | celb, chuA, fyuA, gad, iha, irp2, iss, iucC, iutA, kpsE, kpsII, papA, sat, senB, sitA, traT, usp, yfcV (20) |
| | 131877[e] | 1/5 | | | bla_TEM-1b | | | cba, chuA, cma, fyuA, gad, iha, irp2, iss, iucC, iutA, kpsE, kpsII, papA, sat, sitA, traT, usp, yfcV (20) |
| | 131879 | 2/2 | AMC, CAZ | | bla_CMY-2 | | Col156, IncB/O/K/Z | chuA, fyuA, gad, iha, irp2, iss, iucC, iutA, kpsII, papA, sat, senB, sitA, traT, yfcV (17) |
| | 131905[e] | 1/4 | CAZ, FEP, TSX | aadA5 | bla_CTX-M-27 | aph(3'')-Ib, aph(6)-Id, sul1, sul2, tetA | | chuA, fyuA, gad, iha, irp2, iss, iucC, iutA, kpsE, kpsII, papA, sat, senB, sitA, traT, usp, yfcV (19) |
| | 131906[e] | 2/12 | CAZ, FEP, TSX | aadA5, drf | bla_CTX-M-27 | aph(3'')-Ib, aph(6)-Id, sul1, sul2, tetA | | chuA, fyuA, gad, iha, irp2, iss, iucC, iutA, kpsE, kpsII, papA, sat, senB, sitA, usp (17) |
| | 132351 | 1/3 | AMC, PTZ, CAZ, FEP, GEN, TOB | | bla_CTX-M-15, bla_CMY-2, bla_OXA-1 | aac(6)-Ib-cr, aac(3)-IIa, tetA | Col156, IncB/O/K/Z | chuA, fyuA, gad, iha, irp2, iss, iucC, iutA, kpsII, papA, sat, senB, sitA, traT, usp, yfcV (18) |
| | 162872[d,e] | 4/8 | GEN, TOB, TSX | aadA5, drf | bla_TEM-1b | aph(3'')-Ib, aac(3)-IId, qnrS1, sul1, sul2, tetA | IncI1, IncN, Col156 | chuA, cia, fyuA, gad, iha, irp2, iss, iucC, iutA, kpsE, kpsII, nfaE, papA, sat, senB, sitA, traT, usp, yfcV (21) |
| | 162874 | 1/1 | AMC, PTZ, GEN, TOB, TSX | aadA5, drf | | aph(3'')-Ib, aph(6)-Id, aac(3)-IId, qnrS1, sul1, sul2, tetA | IncI1, IncI2, IncN | chuA, cia, fyuA, gad, iha, irp2, iss, iucC, iutA, kpsE, kpsII, papA, sat, senB, sitA, traT, usp, yfcV (20) |
| | 162878 | 1/1 | TOB | | bla_TEM-1b | | IncI1 | cba, chuA, cia, cma, fyuA, gad, iha, irp2, iucC, iutA, kpsE, kpsII, papA, sat, traT, usp, yfcV (19) |
| | 162879 | 1/1 | | | | | IncI1, Col156 | chuA, cia, fyuA, gad, iha, irp2, iss, iucC, iutA, kpsE, kpsII, papA, sat, sitA, traT, usp, yfcV (18) |
| | 162880[e] | 1/3 | CAZ, FEP, TSX | aadA5, drf | bla_CTX-M-27 | aph(3'')-Ib, aph(6)-Id, sul1, sul2, tetA | Col156 | celb, chuA, fyuA, gad, iha, irp2, iss, iucC, iutA, kpsE, kpsII, papA, sat, senB, sitA, usp, yfcV (19) |

**TABLE 2** (Continued)

| Clade | Lineage according to cgST | No. of patients/ isolates | Resistance profile[a] | Pathogenicity islands | Target β-lactamase genes | Other resistance genes[b] | Plasmids[c] (other than IncF) | Virulence genes[c] (virulence score) |
|---|---|---|---|---|---|---|---|---|
| | 162881[e] | 2/6 | CAZ, FEP, TSX | aadA5, drf | bla_CTX-M-27 | aph(6)-Id, sul1, sul2, tetA | Col156 | celb, chuA, fyuA, gad, iha, irp2, iss, iucC, iutA, kpsE, kpsII, papA, sat, senB, sitA, traT, usp, yfcV (20) |
| | 162883 | 1/1 | AMC, PTZ, CAZ, FEP, GEN, TOB | | bla_TEM-2, bla_SHV-12 | aac(3)-IId | IncI1, IncN | chuA, fyuA, gad, iha, irp2, iss, iucC, iutA, kpsE, kpsII, papA, sat, papA, sat, senB, sitA, usp, yfcV (20) |
| | 162884 | 2/5 | AMC | | | | Col156, IncX1, IncX4 | chuA, cia, fyuA, gad, iha, irp2, iss, iucC, iutA, kpsE, papA, sat, senB, sitA, traT, usp, yfcV (19) |
| | 162886 | 2/2 | CAZ, FEP, TSX | | bla_TEM-1b | aph(3'')-Ib, sul1, sul2 | IncB/O/K/Z | celb, chuA, fyuA, gad, iha, irp2, iss, iucC, iutA, kpsE, kpsII, nfaE, papA, sat, senB, sitA, traT (19) |
| | 162887 | 1/1 | | | bla_TEM-1b | | IncX4 | chuA, fyuA, gad, iha, iha, irp2, iss, iucC, iutA, kpsE, kpsII, papA, sat, sitA, traT, usp, yfcV (19) |
| | 162888[e] | 3/3 | CAZ, FEP, TOB, AK, TSX | drf | bla_CTX-M-14 | aph(3'')-Ib, aac(6')-Ib-cr, sul1, sul2, tetA | IncN | gad, iss, kpsE, kpsII, papA, senB, sitA, yfcV (10) |
| | 162893 | 1/1 | | | | | Col156 | celb, chuA, fyuA, gad, iha, irp2, iss, iucC, iutA, kpsE, kpsII, papA, sat, sitA, usp, yfcV (18) |
| | 162894 | 1/2 | | | bla_TEM-1b | aph(3'')-Ib, aph(6)-Id, sul2 | Col156, IncB/O/K/Z | celb, chuA, fyuA, gad, iha, irp2, iss, iucC, iutA, kpsE, kpsII, papA, sat, senB, sitA, traT, usp, yfcV (20) |
| | 170594 | 1/3 | | | bla_TEM-1b | tetA | | chuA, gad, iha, iss, iucC, kpsE, kpsII, papA, sat, senB, sitA, traT, usp, yfcV (16) |
| C2 | 131563[d,e] | 9/15 | AMC, CAZ, FEP, TOB, TSX | aadA5, drf | bla_CTX-M-15, bla_OXA-1 | aac(6')-Ib-cr, sul1 | | afaA, afaC, afaD, celb, chuA, fyuA, gad, iha, irp2, iss, iucC, iutA, kpsE, kpsII, nfaE, papA, sat, sitA, usp, yfcV (22) |
| | 131564 | 1/1 | AMC, PTZ, CAZ, FEP, TOB, TSX | aadA5, drf | bla_CTX-M-15, bla_OXA-1 | sul1, tetA | | afaA, afaC, afaD, chuA, fyuA, gad, iha, irp2, iss, iucC, iutA, kpsE, kpsII, nfaE, papA, sat, sitA, usp, yfcV (21) |
| | 131872 | 1/2 | | | | | | afaA, afaC, chuA, fyuA, gad, iha, irp2, iss, iucC, iutA, kpsE, kpsII, nfaE, papA, sat, sitA, usp, yfcV (20) |
| | 131876[e] | 1/5 | | drf | | aac(6')-Ib-cr, qnrS1 | IncN | afaA, afaC, afaD celb, chuA, fyuA, gad, iha, irp2, iss, iucC, iutA, kpsE, kpsII, nfaE, sat, sitA, usp, yfcV (21) |
| | 131904[d,e] | 6/25 | AMC, GEN, TOB, TSX | aadA5, drf | bla_OXA-1 | aph(3'')-Ib, aac(6')-Ib-cr, sul1, sul2, tetA | Col156 | chuA, fyuA, gad, hra, iha, ireA, irp2, iss, iucC, iutA, kpsE, kpsII, papA, sat, senB, sitA, traT, usp, yfcV (21) |
| | 131907[e] | 2/6 | | | | tetA | | chuA, fyuA, gad, hra, iha, ireA, irp2, iss, iucC, iutA, kpsE, kpsII, papA, sat, senB, sitA, traT, usp, yfcV (21) |

**TABLE 2** (Continued)

| Clade | Lineage according to cgST | No. of patients/ isolates[a] | Resistance profile[a] | Pathogenicity islands | Target β-lactamase genes | Other resistance genes[b] | Plasmids[c] (other than IncF) | Virulence genes[c] (virulence score) |
|---|---|---|---|---|---|---|---|---|
| | 162873 | 1/1 | | | | tetA | | afaA, afaC, afaD, chuA, fyuA, gad, iha, irp2, iss, iucC, iutA, kpsE, kpsII, nfaE, papA, sat, senB, sitA, traT, usp, yfcV (23) |
| | 162876 | 1/3 | AMC, CAZ, FEP, TSX | aadA5, drf | bla$_{CTX-M-15}$, bla$_{OXA-1}$ | aac(6')-Ib-cr, sul1, tetA | | afaA, afaC, afaD, chuA, fyuA, gad, iha, irp2, iss, iutA, kpsE, kpsII, papA, sat, sitA, traT, usp, yfcV (20) |
| | 162892 | 3/5 | AMC, GEN, TOB, FOS | | bla$_{OXA-1}$ | aac(6')-Ib-cr, aac(3)-IIa, tetA | | chuA, fyuA, gad, iha, ireA, irp2, iss, iucC, iutA, kpsE, kpsII, papA, sat, senB, sitA, traT, usp, yfcV (20) |
| | 162895 | 2/4 | | | | tetA | – | chuA, fyuA, gad, iha, irp2, iss, iucC, iutA, kpsE, kpsII, papA, sat, senB, sitA, traT, usp, yfcV (20) |

[a]Antimicrobial tested: amoxicillin-clavulanic acid (AMC); piperacillin-tazobactam (PTZ); ceftazidime (CAZ); cefepime (FEP); gentamicin (GEN); tobramycin (TOB); amikacin (AK); trimethoprim-sulfamethoxazole (TSX); fosfomycin (FOS).

[b]All isolates harbored gyrA(S83L+D87N) and parC, and all but cgST162895 harbored parE. All isolates also harbored ompT and terC.

[c]IncF plasmids were present in all isolates.

[d]Highly transmitted lineage.

[e]Long colonizers.

**TABLE 3** Features of patients who did and did not acquire a new FQ ST131 *E. coli* lineage

| Variable | Any new acquisition (*n* = 34) | No new acquisition (*n* = 43) | Bivariate *P* value[a] | Adjusted HR (95% CI)[b] | Adjusted *P* value |
|---|---|---|---|---|---|
| Median age in yrs (IQR) | 84 (75 to 88) | 81 (69 to 87) | 0.3 | | |
| Female gender | 21 (61.8) | 29 (67.4) | 0.6 | | |
| Roommate of a colonized resident | 10 (29.4) | 2 (4.7) | 0.003[c] | 3.93 (1.78 to 8.60) | 0.001 |
| Share bathroom | 12 (35.3) | 15 (34.9) | 0.9 | | |
| Use of wheel chair | 13 (38.2) | 14 (32.6) | 0.6 | | |
| Need assistance for hygiene | 32 (94.1) | 40 (93.0) | 1.0[c] | | |
| Need assistance for toilet use | 32 (94.1) | 39 (94.1) | 0.6[c] | | |
| Urinary and/or fecal incontinence | 18 (52.9) | 11 (25.6) | 0.01 | 2.61 (1.31 to 5.20) | 0.006 |
| Urinary catheter | 3 (8.8) | 2 (4.7) | 0.6[c] | | |
| Major surgery (previous yr) | 6 (17.6) | 3 (7.0) | 0.1[c] | | |
| Diabetes mellitus | 7 (20.6) | 7 (16.3) | 0.6 | | |
| Chronic pulmonary disease | 5 (14.7) | 4 (9.3) | 0.4[c] | | |
| Chronic peripheral vascular disease | 28 (82.4) | 27 (62.8) | 0.05 | | |
| Chronic renal disease | 7 (20.6) | 4 (9.3) | 0.1[c] | 2.56 (1.06 to 6.18) | 0.03 |
| Chronic liver disease | 2 (5.9) | 6 (14.0) | 0.2[c] | | |
| Hemiplegia | 5 (14.7) | 3 (7.0) | 0.4[c] | | |
| Malignancy | 3 (8.8) | 4 (9.3) | 1.0[c] | | |
| Unable to provide informed consent | 24 (70.6) | 24 (55.8) | 0.1 | | |
| Prior hospitalization | 6 (17.6) | 7 (16.3) | 0.8 | | |
| Previous antibiotic use | 19 (55.9) | 16 (37.2) | 0.1 | | |

[a]Chi squared except where specified.
[b]Logistic regression; time at risk is included in the model.
[c]Fisher test.

including the four highly transmitted strains. The average duration of colonization in these residents was ≥27.2 weeks (SD 0.6). Colonization in 12 long colonizers (57.1%) was intermittent. C1 and C2 lineages accounted for 10 and four long colonizers, respectively. No significant differences were found between long colonizers and other lineages in phenotypic or genotypic features.

Overall, FQ-R ST131 *E. coli* was isolated from 22 environmental samples: nine from toilets, six from sinks, four from shower heads, one from a chair within a shower, and two from water obtained from the casket drain. Overall, 12 lineages were found in these samples, eight belonging to C1 and two to C2 (two were ESBL-producers). Seven (five C1 and two C2) were also found colonizing residents. Finally, three lineages (one C1 and two C2; none were ESBL producers) were considered to have a clear link with colonized five residents as they were isolated from environmental samples in the bathroom used by these residents.

**Risk factors for acquisition of colonization.** The 34 residents who acquired any new strain of FQ-R ST131 *E. coli* were compared with the 42 who did not (Table 3); the other five patients did not have at least two samples. Among the 34 residents who acquired any new lineage, 23 were from LTCF A and 11 from LTCF B (45.1% and 42.3% of participants from each; *P* = 0.81). Having a colonized roommate, urinary or fecal incontinence, and chronic renal disease were found to be associated with increased risk of acquisition (Table 3). When the analysis was restricted to residents who acquired a C1 lineage, having a colonized roommate (HR = 4.21; 95% CI = 1.71 to 10.36; *P* = 0.002) and urinary or fecal incontinence (HR = 2.82; 95% CI = 1.21 to 6.56; *P* = 0.01) were associated with acquisition.

## DISCUSSION

In this study, we characterized the clinical and molecular epidemiology of FQ-R ST131 *E. coli* in highly dependent patients admitted to two LTCF. We found a high prevalence of intestinal colonization. As the study was done prospectively with repeated sampling over 10 months, we could also estimate the incidence density for the acquisition of these strains, and found that colonization was frequently prolonged,

and identified some risk factors for the acquisition. Specifically, we could delineate some differences in the epidemiology of lineages belonging to C1 and C2 subclades.

Nursing homes were soon detected as reservoirs for ESBL-producing ST131 isolates (9). However, there is scarce data for the burden of colonization with FQ-R ST131 in LTCF. Burgess et al. found 16% and 44% prevalence of colonization in 2 LTCF in Minnesota (10); all ST131 belonged to the subclone H30, and 28% were putative ESBL-producers. Ismail et al. studied five nursing homes in Michigan and provided data on colonization with FQ-R *E. coli* (62.2% were ST131) (11); the overall prevalence of colonization was 21.6% but with a range of 7.6% to 52.6%, and the incidence was 1.05 cases per 1,000 patient days. The data in our study, performed in LTCF caring for highly dependent residents, were in line with those results. Therefore, this study confirms that LTCF are important reservoirs for these organisms. However, we provided additional information by reporting a longitudinal assessment of colonization status over a long period of time, which allowed us to evaluate the rate of transmission events.

Lineages belonging to C1 and C2 showed some subtle differences, already characterized in other studies: some resistant determinants such as *aac6'-ib-cr* and *bla*$_{OXA-1}$, and some virulence traits were more frequent among C2 lineages. In fact, a higher virulence score was previously described in C2 isolates, but the specific genes associated with C2 instead of C1 differed between studies (12, 13). Our data are insufficient to characterize whether the virulence genes associated with C2 isolates in our study are associated with any specific epidemiological behavior. While the proportion of C1 and C2 strains producing ESBL were similar, CTX-M-27 was only found in some C1 lineages, while CTX-M-15 was associated with C2 isolates, as expected. Colonization by C1 strains producing CTX-M-27 was more frequent than C2 producing CTX-M-15 in two geriatric rehabilitation wards in Tel Aviv in 2012 (14). We also found that the pooled prevalence and incidence of new acquisitions were somehow higher for lineages belonging to subclade C1 than to subclade C2 even if not producing ESBLs.

The fact that highly transmitted lineages did not show different phenotypic or genotypic features compared with other lineages suggest that the studied microbiological factors are not relevant for their transmissibility, but it does not discard a potential implication of other unknown factors. Also, the lack of association may just reflect a lack of statistical power. Alternatively, the results of the risk factors analysis suggest that patients-related factors facilitating cross-transmission may be more relevant. In addition, environmental colonization might also play a role in some cases.

Intestinal colonization with ESBL-producing and FQ-R ST131 *E. coli* has been found to be more prolonged than with other STs in LTCF residents (11, 15). We found that a substantial proportion C lineages were able to cause prolonged colonizations. Again, no specific microbiological factors among those studied would be associated with longer colonization.

Most studies on risk factors for colonization with ST131 in LTCF residents has been performed in ESBL-producers. We found three previous studies in FQ-R isolates. Han et al. studied FQ-R *E. coli* colonization in three LTCF in Pennsylvania and found fecal incontinence, receipt of amoxicillin-clavulanate, and presence of a urinary catheter to be independent risk factors (16); 78% of the isolates were ST131 (17). Ismail et al. found lower exposure to enteral feeding tube and higher to urinary catheter among patients colonized, but multivariate analysis was not performed (11). Burgess et al. studied prevalent cases colonized by FQ-R ST131 isolates; inability to sign consent and presence of a decubitus ulcer were found as risk factors for colonization (10). In this study, sharing room with a previously colonized resident and urinary or fecal incontinence were risk factors, which would be explained in terms of facilitating cross-transmission. We could not find that antibiotic use increased the risk. In a previous study, we did not find antibiotic use to be a risk factor for colonization with any ST131 (around 70% were FQ-R) in households or hospital wards (18). Therefore, although antibiotics might facilitate the acquisition by altering the protective microbiota, this might be less relevant than cross transmission.

This study has limitations which must be considered when interpreting the results. First, it was performed in a specific epidemiological setting so the results may not be applicable to other situations. Second, we may have missed some colonization events since the sensitivity of fecal/rectal swab is limited. Third, we may have not identified some risk factors because of lack of statistical power. Finally, we could not perform WGS for all isolates which might have provided more robust data on transmission events.

In conclusion, residents in high-risk LTCF are important reservoirs for FQ-R ST131; C1 lineages are emerging. The risk factors found suggest that cross transmission was the main mechanism of transmission in this specific epidemiological environment.

## MATERIALS AND METHODS

**Design, location, and study period.** We conducted a prospective observational study in residents from two LTCFs. Both LTCFs, with 92 and 50 residents each, admit mostly elderly dependent persons for basic activities in different degrees, and are similar in the care provided and resources. All residents admitted at the date of the study start were eligible.

Fecal swabs (or rectal when fecal samples were not available) were obtained from the participants over two 12-week periods from March to May 2018, and from November 2018 to January 2019. Summer sampling was avoided because some residents temporarily leave the LTCF. In each period, the samples were obtained in visits performed at baseline (week 0), and at weeks 1, 4, and 12. Therefore, the visits and samples are labeled as weeks 0, 1, 4, 12, 28, 29, 32, and 40.

The project was approved by the Ethic Committee of Hospital Universitario Virgen Macarena, and written informed consent was obtained from the residents or their relatives. In the absence of the later, the ethics committee waived the need to obtain a consent.

**Variables and definitions.** A resident was considered colonized when a FQ-R ST131 *E. coli* was isolated from a fecal or rectal swab. The duration of colonization by each FQ-R ST131 *E. coli* lineage was studied only for those with at least 12 weeks of follow-up, and was defined as prolonged when it was ≥12 weeks. Colonization was considered intermittently detected when there were negative samples between positive ones. Epidemiological and clinical data were collected at baseline and during the different visits. The variables are shown in Table 1.

**Microbiological studies.** Fecal or rectal swabs were taken using FecalSwabs (Copan) and environmental samples were taken from every toilet surface, sink, or shower head using ESwab(Copan). All samples were enriched in broth for 18 h and inoculated on chromogenic UTI-agar (Oxoid) with 2 mg/L ciprofloxacin. All morphologically different colonies of *E. coli* isolated were selected for studied. Screening for ST131 was performed by PCR using primers for O25b *rfb*, allele 3 of *pabB* gene, and phylogroup B$_{23}$ typing (6). All ST131 positive isolates were further studied. The antimicrobial susceptibility was studied by NMDRM1 Microscan panels (Beckam Coulter). The genetic relatedness of O25b/*pabB3*/B$_{23}$ positive isolates was studied by WGS for all baseline isolates in a previous study (8) and subsequent isolates from the same patient by XbaI PFGE analysis (https://pulsenetinternational.org/). PFGE dendrograms were created with BioNumerics 7.5 software (Applied Maths, bioMérieux), using the Dice coefficient and the unweighted pair group method (UPGM) (position tolerance 1%). Isolates with >1 band difference were assigned to different pulsotypes (19). All new different pulsotypes from baseline ones and those with different susceptibility pattern within a pulsotype were also studied by WGS. Draft genomes were generated using the Nextera Flex DNA sample preparation kit and Illumina MiSeq 2000 reads and were assembled *de novo* with CLC genomics software. The assembled genomes were annotated by using Resfinder 3.2 (20) and CARD (21) for resistance determinants; MLSTFinder 2.0 (22) and FimHTyper 1.0 to assign ST and fimH types, respectively; virulenceFinder for virulence traits and PlasmidFinder to identify plasmid. For the purposes if this article, strains within the complex were defined by using cgMLST V1 (Hiercc) of *Escherichia coli/Shigella* EnteroBase database scheme (https://enterobase.warwick.ac.uk/), and were referred to as strains, cgSTs, or lineages. A BLAST search was performed for the M27PP1 region specific to clade 1 using strain KUN5781 as Matsumura et al. (3).

Transmission was considered when two isolates showed identical cgMLST or pulsotype, together with identical content in terms of resistance determinants, virulence factors, and plasmid replicons. Virulence score was defined by using Dahby's scheme (17, 23).

**Statistical analyses.** The prevalence of colonization with FQ-R ST131 *E. coli* was calculated at each time point. In addition, incidence density was calculated (i) for the first acquisition of FQ-R ST131 *E. coli*, calculated as incident cases per 100 resident weeks and considered until the first detection of colonization or the end of follow-up (censorship); and (ii) for the acquisition of any new cgST of FQ-R ST131 *E. coli*, calculated as incident acquisition of any new lineage per 100 resident weeks. For the latter, all patients were considered at risk for acquiring a new lineage during all their follow-up period.

Bivariate comparisons were performed using chi square or Fisher exact test as appropriate for categorical variables, and the Mann-Whitney U test for continuous variables. A *P* value <0.05 was considered significant, except when multiple comparisons were made, for which <0.01 was requested. The hazards for the first acquisition of a FQ-R ST131 *E. coli* lineage were studied by comparing the time until acquisition or the end of follow-up (censoring) if no acquisition occurred, using Cox regression, after checking the proportionality of hazards. Variables with a univariate *P* value <0.2 were included in the model and selected using a manual stepwise backward process. Collinearity was checked.

**Data availability.** Raw sequencing reads of 64 *E. coli* ST131 isolates (those obtained from patients and environment) were deposited in BioProject PRJNA742861.

## ACKNOWLEDGMENTS

This study was funded by the Instituto de Salud Carlos III, Ministerio de Ciencia, Innovación y Universidades (grants no. AC16/00072 and AC16/00076) within the 2016 Joint Call framework for transnational research projects on the transmission dynamics of antibacterial resistance, JPI-EC-AMR Program (Joint Programming Initiative on Antimicrobial Resistance), European Union.

The authors also received support for the research from the Plan Nacional de I+D+i 2013–2016 and Instituto de Salud Carlos III, Subdirección General de Redes y Centros de Investigación Cooperativa, Ministerio de Ciencia, Innovación y Universidades, Spanish Network for Research in Infectious Diseases (REIPI 315 RD16/0016/0001), cofinanced by the European Development Regional Fund "A Way to Achieve Europe," Operative Program Intelligence Growth 2014–2020.

The funders had no role in design, analysis, or reporting the results.

We thank all participants and their relatives, as well as the staff from the participating LTCFs, especially Rocio Medina and Cristina Belloso.

The authors had no conflicts of interest.

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
