## [Reviewer comments · Microbiology Spectrum]

Microbiology Spectrum

Prevalence, incidence and risk factors for intestinal colonization due to fluoroquinolone-resistant ST131 *Escherichia coli*: a longitudinal study in highly-dependent, long term-care facility residents.

Elena Salamanca Rivera, Lorena López-Cerero, Jose Manuel Rodriguez-Martínez, Alvaro Pascual, and Jesús Rodríguez-Baño

Corresponding Author(s): Jesús Rodríguez-Baño, Hospital Universitario Virgen Macarena

Review Timeline:

Submission Date:	May 6, 2022
Editorial Decision:	May 31, 2022
Revision Received:	July 1, 2022
Accepted:	July 13, 2022

Editor: Ana Paula Carvalho-Assef

Reviewer(s): Disclosure of reviewer identity is with reference to reviewer comments included in decision letter(s). The following individuals involved in review of your submission have agreed to reveal their identity: Cláudio Marcos Rocha-de-Souza (Reviewer #1)

Transaction Report:

DOI: <https://doi.org/10.1128/spectrum.01673-22>

May 31, 2022

Prof. Jesús Rodríguez-Baño
Hospital Universitario Virgen Macarena
Enfermedades Infecciosas
Avda Dr Fedriani 3
Sevilla 41009
Spain

Re: Spectrum01673-22 (Prevalence, incidence and risk factors for intestinal colonization due to fluoroquinolone-resistant ST131 *Escherichia coli*: a longitudinal study in highly-dependent, long term-care facility residents.)

Dear Prof. Jesús Rodríguez-Baño:

Link Not Available

Sincerely,

Ana Paula Carvalho-Assef

Journals Department
Reviewer comments:

Reviewer #1 (Comments for the Author):

The manuscript entitled "Prevalence, incidence and risk factors for intestinal colonization due to fluoroquinolone-resistant ST131 *Escherichia coli*: a longitudinal study in highly dependent, long term-care facility residents." concerns a prospective observational study in residents from long term-care facilities in order to investigate the colonization and epidemiological features of clade C fluoroquinolone-resistant ST131 *E. coli*. As known the ST131 is the most frequently isolated fluoroquinolone-resistant (FQR) *E. coli* clone worldwide and a major cause of urinary tract and bloodstream infections. Overall, the work is well performed and results would be of interest for the journal's readership. However, there are minor few points that needed to be addressed which

in my opinion would improve the manuscript.

a) This is just a question. Why authors only focused on clade C since there are some papers showing the increasing of clade A importance. Clade A ST131 are poorly studied but in terms of lethality are considered equivalent to clade C in a mouse sepsis model of infection. Furthermore, clade A isolates also carry blaCTX-M-27 and dual parC-1aAB and gyrA-1AB fluoroquinolone resistant mutations, a feature previously considered unique to clade C isolates. In my opinion, authors missed out here a great opportunity to enrich the work

b) Page 09: Change 49 residents to 34. The sum does not match

c) Page 09: "regarding virulence genes, afaA, afaC and afa D were less frequent....." Please write the complete name before the acronyms are used for the first time. The same for nfaE gene. Why these virulence factors are so important in ST131 Escherichia coli strains? This information is not contextualised in the text.

d) Page 11: There is no Table 04. "The 34 residents who acquired any new strain of FQ-R ST131 E. coli were compared with the 42 who did not (table 4)"

e) table 2 shows several virulence factor genes. However, in any place in the main text the authors discuss the relevance of these genes in the ability of fluoroquinolone-resistant ST131 E. coli in colonize and spread. Looks that this information was not so explored. The same for Plasmid incompatibility groups.

f) In my opinion, supplementary figure should be included in the main text

Reviewer #2 (Public repository details (Required)):

The WGS data should be deposited in a database.

Reviewer #2 (Comments for the Author):

In the manuscript Spectrum01673-22, the authors traced fluoroquinolone-resistant Escherichia coli ST131 clade C in two long term care facilities. They report that cross-transmission among individuals residing in proximity is more important as a risk factor than microbiological or other virulence factors. The work is interesting, important, and described in detail. The authors also mention the limitations of their data which must be commended. I have following comments.

Major Comments:

1. The authors must emphasize how their work is different in terms of design and interpretations than their cited reference # 14 and 15.
2. The manuscript is largely descriptive. Addition of a diagram which shows incidence and transmission tree with color codes would definitely improve the manuscript. Please see Figure 2 in PMID: 27377746 and Figure 1 in PMID: 34367312 as examples. The authors tried this in supplementary figure but that figure is confusing. They also called the supplementary figure as 'Table 4' in 'Risk factors for acquisition of colonization' section. There is no Table 4 in the manuscript. I suggest that the authors replace this supplementary figure with a transmission tree as a main figure.
3. Since the major claim of the study is cross-transmission through proximity, the authors should also further explain the causal link between transmission isolates and environmental isolates. They do so in the first paragraph of page 11 but, addition of a phylogenetic tree amongst all the isolates will add clarity (they already have the WGS data). Since the number of samples is on the lower side, addition of this data will propel the correlation to causation.
4. In table 3, the author should explain why the correlation with the renal disease is not taken into account. Perhaps, is it because of the n (and hence the confidence) being small?
5. The authors should expand the discussion on the virulence factors they found (table 3).

Minor Comments:

1. The authors should add line numbers in the manuscript.
2. In the abstract, please expand the LTCF abbreviation for the readers.
3. Fix the grammar in line: "was avoided because some residents transitory leave the LTCF."
4. Where were the environmental samples collected from? The authors mention toilets, shower heads, etc. but they should mention the location such as which facility and same or different rooms etc.

Staff Comments:

Preparing Revision Guidelines

Please return the manuscript within 60 days; if you cannot complete the modification within this time period, please contact me. If you do not wish to modify the manuscript and prefer to submit it to another journal, please notify me of your decision immediately so that the manuscript may be formally withdrawn from consideration by Microbiology Spectrum.

The manuscript entitled “Prevalence, incidence and risk factors for intestinal colonization due to fluoroquinolone-resistant ST131 *Escherichia coli*: a longitudinal study in highly dependent, long term-care facility residents.” concerns a prospective observational study in residents from long term-care facilities in order to investigate the colonization and epidemiological features of clade C fluoroquinolone-resistant ST131 *E. coli*. As known the ST131 is the most frequently isolated fluoroquinolone-resistant (FQR) *E. coli* clone worldwide and a major cause of urinary tract and bloodstream infections. Overall, the work is well performed and results would be of interest for the journal’s readership. However, there are minor few points that needed to be addressed which in my opinion would improve the manuscript.

- a) This is just a question. Why authors only focused on clade C since there are some papers showing the increasing of clade A importance. Clade A ST131 are poorly studied but in terms of lethality are considered equivalent to clade C in a mouse sepsis model of infection. Furthermore, clade A isolates also carry *bla*_{CTX-M-27} and dual *parC-1aAB* and *gyrA-1AB* fluoroquinolone resistant mutations, a feature previously considered unique to clade C isolates. In my opinion, authors missed out here a great opportunity to enrich the work
- b) **Page 09**: Change 49 residents to 34. The sum does not match
- c) **Page 09**: “regarding virulence genes, *afaA*, *afaC* and *afaD* were less frequent.....” Please write the complete name before the acronyms are used for the first time. The same for *nfaE* gene. Why these virulence factors are so important in ST131 *Escherichia coli* strains? This information is not contextualized in the text.
- d) **Page 11**: There is no Table 04. “The 34 residents who acquired any new strain of FQ-R ST131 *E. coli* were compared with the 42 who did not (table 4)”
- e) **table 2** shows several virulence factor genes. However, in any place in the main text the authors discuss the relevance of these genes in the ability of fluoroquinolone-resistant ST131 *E. coli* in colonize and spread. Looks that this information was not so explored. The same for Plasmid incompatibility groups.
- f) In my opinion, supplementary figure should be included in the main text

The Editor

Sevilla, 7 June 2022

Dear Editor,

Thank you for providing us the opportunity to submit a revised version of the manuscript entitled "Prevalence, incidence and risk factors for intestinal colonization due to fluoroquinolone-resistant ST131 Escherichia coli: a longitudinal study in highly-dependent, long term-care facility residents."; we also thank the reviewers for their comments which have been really helpful in improving the manuscript.

Please see our answers to your and the reviewers' comments below.

Sincerely,

Jesus Rodriguez-Baño
On behalf of all authors

Reviewers' comments:

Reviewer #1

The manuscript entitled "Prevalence, incidence and risk factors for intestinal colonization due to fluoroquinolone-resistant ST131 Escherichia coli: a longitudinal study in highly dependent, long term-care facility residents." concerns a prospective observational study in residents from long term-care facilities in order to investigate the colonization and epidemiological features of clade C fluoroquinolone-resistant ST131 E. coli. As known the ST131 is the most frequently isolated fluoroquinolone-resistant (FQR) E. coli clone worldwide and a major cause of urinary tract and bloodstream infections. Overall, the work is well performed and results would be of interest for the journal's readership. However, there are minor few points that needed to be addressed which in my opinion would improve the manuscript.

a) This is just a question. Why authors only focused on clade C since there are some papers showing the increasing of clade A importance. Clade A ST131 are poorly studied but in terms of lethality are considered equivalent to clade C in a mouse sepsis model of infection. Furthermore, clade A isolates also carry blaCTX-M-27 and dual parC-1aAB and gyrA-1AB fluoroquinolone resistant mutations, a feature previously considered unique to clade C isolates. In my opinion, authors missed out here a great opportunity to enrich the work

Response: We thank the reviewer for the general comment on our manuscript. Regarding the decision to investigate clade A, we agree with the reviewer about the interest of this clade due to lack of data; however, when the study was designed, we were particularly interested in FQ-resistant clade C and the emergence of CTX-M-27 within.

b) Page 09: Change 49 residents to 34. The sum does not match

Response: We are sorry but we think the data is correct. Please note there were 34 different cgSTs among all participants in which an isolate was found at any point in the study (49 residents were positive at some point in the study but some cgSTs recurred among them).

c) Page 09: "regarding virulence genes, afaA, afaC and afa D were less frequent....." Please write the complete name before the acronyms are used for the first time. The same for nfaE gene. Why these virulence factors are so important in ST131 Escherichia coli strains? This information is not contextualised in the text.

Response: Thank you very much for your suggestion. We guess the reviewer refers to the name of the proteins codified by the genes. We have consulted other manuscripts published by ASM and only the names of the genes are provided. However, we will follow any indication of the Editor in this regard.

We added some sentences in the Discussion to explain the interest of the findings regarding the virulence factor genes, and added two references.

d) Page 11: There is no Table 04. "The 34 residents who acquired any new strain of FQ-R ST131 E. coli were compared with the 42 who did not (table 4)"

Response: Thank you for raising this mistake. We corrected it.

e) table 2 shows several virulence factor genes. However, in any place in the main text the authors discuss the relevance of these genes in the ability of fluoroquinolone-resistant ST131 E. coli in colonize and spread. Looks that this information was not so explored. The same for Plasmid incompatibility groups.

Response: As explained above, we added some sentences in this regard. We have looked at associations between virulence factors genes or plasmids and high transmissibility, but as explained we could not find any association., as stated already in the Discussion section. We also added the possibility that this may be due to lack of statistical power

f) In my opinion, supplementary figure should be included in the main text

Response: Following the reviewer suggestion, we have included the figure in the main text as Figure 1.

Reviewer #2

The WGS data should be deposited in a database.

Response: Of course, the reviewer is right. We have deposited the WGS in a database and added in the manuscript: Raw sequencing reads of 64 E. coli ST131 isolates (those obtained from patients and environment) were deposited in BioProject PRJNA742861.

In the manuscript Spectrum01673-22, the authors traced fluoroquinolone-resistant Escherichia coli ST131 clade C in two long term care facilities. They report that cross-transmission among individuals residing in proximity is more important as a risk factor than microbiological or other virulence factors. The work is interesting, important, and described in detail. The authors also

mention the limitations of their data which must be commended. I have following comments.

Major Comments:

1. The authors must emphasize how their work is different in terms of design and interpretations than their cited reference # 14 and 15.

Response: Thank you for raising this important aspect. We added a sentence in the Discussion; our study provided longitudinal data allowing an assessment of transmission events while references 14, 15 and 16, while very interesting, provided point prevalence data.

2. The manuscript is largely descriptive. Addition of a diagram which shows incidence and transmission tree with color codes would definitely improve the manuscript. Please see Figure 2 in PMID: 27377746 and Figure 1 in PMID: 34367312 as examples.

The authors tried this in supplementary figure but that figure is confusing. They also called the supplementary figure as 'Table 4' in 'Risk factors for acquisition of colonization' section. There is no Table 4 in the manuscript.

I suggest that the authors replace this supplementary figure with a transmission tree as a main figure.

Response: Thank you for the suggestion. Due to the high number of participants, samples and clones, over a long study period, we have considered different ways to represent the transmission events together with duration of colonization. In fact, we had considered using a similar figure to those suggested by the reviewer but we think these (the first refers to transmission of Ebola and the second mostly to viruses) are not the most appropriate for the information we needed to provide. In fact, these figures don't capture some essential aspects such as the duration of colonization and the time elapsed among transmission events. However, if the Editor suggests us otherwise we would provide a different figure.

Indeed, there is an error when referencing that Table/Figure, we corrected it, and as suggested, we added it as a figure within the main document. We are sorry for this mistake.

3. Since the major claim of the study is cross-transmission through proximity, the authors should also further explain the causal link between transmission isolates and environmental isolates. They do so in the first paragraph of page 11 but, addition of a phylogenetic tree amongst all the isolates will add clarity (they already have the WGS data). Since the number of samples is on the lower side, addition of this data will propel the correlation to causation.

Response: We agree with the reviewer that providing a phylogenetic tree of all isolates would be of interest. However, due to funds limitation and as explained in the text, we could perform WGS only for specific representatives in each pulsotype plus and, in addition, for those with different susceptibility phenotypes within each pulsotype. We added a sentence with this limitation in the Discussion section.

4. In table 3, the author should explain why the correlation with the renal disease is not taken into account. Perhaps, is it because of the n (and hence the confidence) being small?

Response: We are not sure we understand the comment; chronic renal disease was found to be associated in adjusted analysis with new acquisitions; the fact that it was not significant to a p value <0.05 in the bivariate analysis means that this bivariate association was confounded by other variables. This is the reason why it is good practice to consider including variables with a higher p value in multivariate analysis. We hope to have clarified what the reviewer meant.

5. The authors should expand the discussion on the virulence factors they found (table 3).

Response: we did so, please see our answer to reviewer #1 in this regard.

Minor Comments:

1. The authors should add line numbers in the manuscript.

Response: We did as suggested

2. In the abstract, please expand the LTCF abbreviation for the readers.

Response: We did as suggested

3. Fix the grammar in line: "was avoided because some residents transitory leave the LTCF."

Response: We did as suggested

4. Where were the environmental samples collected from? The authors mention toilets, shower heads, etc. but they should mention the location such as which facility and same or different rooms etc.

Response: We sampled all existing sinks, toilets, showers, etc. in both residences. We added this information.

July 13, 2022

Prof. Jesús Rodríguez-Baño
Hospital Universitario Virgen Macarena
Enfermedades Infecciosas
Avda Dr Fedriani 3
Sevilla 41009
Spain

Re: Spectrum01673-22R1 (Prevalence, incidence and risk factors for intestinal colonization due to fluoroquinolone-resistant ST131 *Escherichia coli*: a longitudinal study in highly-dependent, long term-care facility residents.)

Dear Prof. Jesús Rodríguez-Baño:

Your manuscript has been accepted, and I am forwarding it to the ASM Journals Department for publication. You will be notified when your proofs are ready to be viewed.

Sincerely,

Ana Paula Carvalho-Assef
Editor, Microbiology Spectrum
